# Aquaporins Are Differentially Regulated in Canine Cryptorchid Efferent Ductules and Epididymis

**DOI:** 10.3390/ani11061539

**Published:** 2021-05-25

**Authors:** Caterina Squillacioti, Nicola Mirabella, Giovanna Liguori, Giuseppe Germano, Alessandra Pelagalli

**Affiliations:** 1Department of Veterinary Medicine and Animal Productions, University of Napoli Federico II, 80100 Naples, Italy; caterina.squillacioti@unina.it (C.S.); nicola.mirabella@unina.it (N.M.); giuseppe.germano@unina.it (G.G.); 2Department of Advanced Biomedical Sciences, University of Napoli Federico II, 80100 Naples, Italy; alessandra.pelagalli@unina.it; 3Institute of Biostructures and Bioimages, National Research Council, 80145 Naples, Italy

**Keywords:** aquaporins, epididymis, dog, cryptorchidism, homeostasis, male reproduction

## Abstract

**Simple Summary:**

The distribution and expression of aquaporins (AQPs) in the testes and spermatozoa of several animal species play important roles in spermatogenesis and spermatozoon transit in this region. The aim of this study was to evaluate AQP7, AQP8, and AQP9 localization and expression in the efferent ductules and epididymal regions (the caput, corpus, and cauda) of normal and cryptorchid dogs. The results from immunohistochemistry, Western blotting, and real-time reverse transcription polymerase chain reaction (RT-PCR) show regional tissue distributions, particularly at the level of the epithelium of efferent ductules and both the regions caput and cauda of the canine cryptorchid epididymis. These findings support the hypothesis that these channel proteins respond differently to multiple stimuli that cause cryptorchidism (hormones, heat, osmolarity, etc.) and participate in the mechanisms of cell “resilience” or apoptosis taking place in the epididymis.

**Abstract:**

The efferent ductules and the epididymis are parts of the male reproductive system where spermatozoa mature. Specialized epithelial cells in these ducts contribute to the transport of fluids produced by spermatozoa’s metabolic activity. Aquaporins (AQPs) have been demonstrated to be expressed in the spermatozoan membrane and testis epithelial cells, where they contribute to regulating spermatozoan volume and transit through environments of differing osmolality. Due to the lack of detailed literature regarding AQP expression in the canine male genital tract, the aim of this study was to investigate both the distribution and expression of AQP7, AQP8, and AQP9 in the efferent ductules and epididymal regions (caput, corpus, and cauda) from normal and cryptorchid dogs by using immunohistochemistry, Western blotting, and real-time reverse transcription polymerase chain reaction (RT-PCR). Our results show different patterns for the distribution and expression of the examined AQPs, with particular evidence of their upregulation in the caput and downregulation in the cauda region of the canine cryptorchid epididymis. These findings are associated with a modulation of Hsp70 and caspase-3 expression, suggesting the participation of AQPs in the luminal microenvironment modifications that are peculiar characteristics of this pathophysiological condition.

## 1. Introduction

The epididymis is a tract in the male reproductive system that plays a pivotal role in the complex processes of sperm maturation, transit, and storage. It is a convolute tubule divided into three anatomical parts—the caput, corpus, and cauda—representing physiologically unique compartments that create the luminal microenvironment essential for sperm survival. Of note, specialized epithelial cells in the epididymal segments contribute to the metabolic activity of spermatozoa during its transit along the epididymis. In detail, the proximal regions of the epididymis are devoted to sperm maturation [1], while sperm storage takes place in the distal region [2].

In humans and larger mammals, but not in rodents, a major portion of the proximal region of the epididymis is characterized exclusively by efferent ductules, a complex histological structure localized particularly at the junction of the caput epididymis. Efferent ductules are channels involved in sperm transit from the rete testis to the epididymis. These ducts have been described as compact and small if compared with the epididymal tubules, which appear long and convoluted [3]. In rodents, the epididymis is divided into four regions: initial segment, caput, corpus, and cauda, whereas in large animals and humans, the initial segment is absent [3]. As described in humans, and also in dogs, the proximal region of epididymis is occupied by blind-ending efferent ductules [3,4,5]. A crucial role of these aforementioned tubules is to increase sperm concentration by a process of water reabsorption (up to 96% of the testicular fluid) prior to reaching the epididymis [6]. This event is also characterized by active transport of ions, followed by passive movements of water under estrogen regulation [3,7,8].

Studies have demonstrated that the epididymis is a metabolically active tissue comprising morphologically and functionally distinct cells that regulate important processes, including fluid reabsorption, secretion of proteins, and the production of antioxidants [9]. The luminal water reabsorption by the epididymal epithelium can have the following three main consequences [10]: (1) an increase in the sperm concentration in the deferent ducts of the testes, (2) the modulation of protein concentrations [11,12], and (3) an increase in osmolality that facilitates the regulation of sperm volume [13].

To realize these processes, the epididymal epithelial cells of these ducts are specialized, as demonstrated by a wide repertoire of protein and gene expression, transporters, and receptors [14,15] that allow them to respond uniquely to different stimuli, such as hormonal and other regulatory factors. Temperature is another major factor influencing epididymal physiology [2]. Some studies have demonstrated that changes in body temperature modify the ionic and protein composition of the cauda fluid by altering the cauda epithelium and, thus, impair its normal function of storing and prolonging the life of spermatozoa [16,17]. Conditions including cryptorchidism, characterized by modifications of testicular temperature caused by the failure of one or both testes to descend into the scrotum, and a varicocele, which is an enlargement of venous blood vessels within the scrotum, have been demonstrated to be detrimental to sperm production and viability [18,19]. Moreover, heat stress during cryptorchidism induces male germ cell death by apoptosis, which is considered to be the predominant mechanism involved in this condition [20]. It has been well demonstrated that heat stress, altering the normal cellular functions, induces the synthesis of several heat shock proteins, including Hsp70. This protein has a critical role as a marker protein of cell thermotolerance [21], and its expression has been found dramatically increased in rat epididymis as a result of the exposition of animals to acute heat stress (39 °C for 0.5, 1, and 3 h) [22]. More recently, similar results were also observed in male cavy exposed for 60 days to 39 °C, demonstrating decreased sperm mobility, sperm count, and testicular antioxidant enzymes accompanied by an increased serum level of Hsp40 [23].

In the last decade, studies focusing on the mechanisms involved in spermatozoon maturation and the luminal microenvironment in contact with spermatozoa have demonstrated the presence of specific water channel proteins named aquaporins (AQPs), expressed either in the spermatozoan membrane or in the testis epithelial cells. The results have suggested that these water channel proteins participate in processes involving water, ions, and molecules, such as lactate in sperm maturation [24,25,26]. AQPs belong to a family of 13 integral membrane proteins normally expressed in the tissues of all living organisms that play a fundamental and pivotal role in water and small-solute transport across cell membranes. AQPs are generally divided into two main groups based on their transfer specificity: classical water-transporting AQPs (AQP0, 1, 2, 4, 5, 6, and 8) and solute-transporting aquaglyceroporins (AQP3, 7, 9, and 10). In addition, a third aquaporin group, comprising AQP11 and AQP12, the most distantly related paralogs, are defined as superaquaporins [27] or subcellular aquaporins [28], according to their cellular localization.

Characterization studies on AQPs in the male reproductive tracts of several domestic animals (buffalo, hamsters, horses, pigs, rams, and sheep) [29,30,31,32,33,34,35] have demonstrated the presence of these channel proteins in different tracts, also showing different functions according to the examined regions, cell types, membrane domains, species, breeding seasons, and age of the subjects [36,37,38,39].

Regarding domestic animal species, studies on dogs have often been conducted regarding either testicular organization and function [40,41,42] or clinical problems related to the specific dysfunction of this tissue [43]. From the limited studies reported, data on the expression and possible role of AQPs in the canine male reproductive tract are limited to the testicular area [38,39,44]; the presence of AQP1 and AQP9 in different cell types in the testis [38,39,44]; and the presence of AQP3, 4, 7, and 9 at the level of the gubernaculum testis (GT) [45], suggesting the need for the further investigation of their roles using disease models of the reproductive system.

The available data, mostly obtained from laboratory animal studies, demonstrate that systematic analyses of the expression of AQPs in epididymal epithelial cells have been lacking [46] and that there have been few studies focused on AQP8 [37,47]. Noteworthily, as indicated by Kirchhoff (2002) [48], the canine epididymis represents a useful model with high relevance to humans, especially at the level of tissue-specific gene expression. Comparable results at the molecular level have been reported from analyzing epididymis from a wide variety of dog breeds and mongrels [49], thus emphasizing the importance of studies on this topic.

The aim of this study was to investigate both the distribution and expression of AQP7, 8, and 9 in the caput, corpus, and cauda of the epididymis of normal and cryptorchid dogs by using real-time reverse transcription polymerase chain reaction (RT-PCR), Western blotting, and immunohistochemistry. In the second part of the study, we aimed at analyzing, in the same tracts of normal and cryptorchid canine epididymis, Hsp70 and caspase-3 expression by Western blotting, considering their possible relation to AQPs since channel proteins also regulate cell homeostasis in relation to a physiological process leading to cell death, apoptosis, which is a common characteristic of cryptorchidism. 

## 2. Materials and Methods

### 2.1. Animals and Tissue Collection 

A total of 10 dogs (5 normal and 5 cryptorchid with unilateral cryptorchidism), medium sized, and aged between two and eight years old, were enrolled in this study. The dogs came to the University of Federico II veterinary clinic, Naples, Italy, for orchiectomy between April and November 2018. All the experimental procedures were conducted with the approval of the University of Naples Federico II and local Ethics Committee (approval number 0-050-377), in accordance with relevant national and international guidelines. Epididymis samples were obtained immediately after the removal of testicles and transported on ice (+4 °C) within 40 min for further processing. The epididymal tissues were divided into two groups: normal epididymis (epididymis from normal dogs) and cryptic epididymis (retained epididymis from cryptorchid dogs). From each group, the caput, corpus, and cauda of the epididymis were sectioned and fixed in Bouin’s fluid for immunohistochemistry. For Western blotting and real-time RT-PCR, the samples were frozen at −80 °C until analysis.

### 2.2. Immunohistochemistry

Fixed samples were embedded in paraffin and sectioned at a thickness of 6 μm. For immunohistological study, the sections were deparaffinized and hydrated through xylene in a graded ethanol series. They were also pretreated with citrate buffer (pH 6) for 2 min (two times) in a microwave (700 watts) for antigen retrieval. After cooling for 30 min, the samples were incubated in 3% H_2_O_2_ for 20 min, in a humid chamber, to quench endogenous peroxidase. Then, they were washed in phosphate-buffered saline (PBS) for 5 min (three times) and blocked in diluted normal goat serum (Vector laboratories, Inc., Burlingame, CA, USA) for 30 min. Next, they were incubated with rabbit polyclonal antibodies against AQP7 (orb13253), AQP8 (orb101163), and AQP9 (orb10127) (dil., 1:500; Biorbyt LLC, San Francisco, California, USA), at 4 °C overnight.

The following day, the sections were washed with PBS and incubated with a secondary antibody, Ultra-Polymer Goat anti-Rabbit/Mouse IgG (ImmunoReagents, Raleigh, NC, USA) conjugated with a peroxidase polymer backbone (1:4), for 30 min in a humid chamber. After washes with PBS, they were treated with diaminobenzidine 3,3′ tetra hydrochloride (DAB) (Vector laboratories, Burlingame, CA, USA) until the desired stain intensity developed. Finally, the sections were dehydrated through an ascending alcohol series mounted with Eukitt**^®^** (Sigma-Aldrich, Taufkirchen, Germany), observed under a Nikon Eclipse E 600 light microscope, and photographed using a Nikon Coolpix 8400 digital camera. The negative controls were reactions performed without the primary antibody incubation step, as described elsewhere [50], and a sample from a rat kidney was used as the positive control.

### 2.3. Western Blotting

The epididymal tissues were processed for Western blotting according to the previous adopted method [51]. In brief, the tissues were homogenized in RIPA buffer and then centrifuged at 14,000 rpm for 15 min, at 4 °C, to remove the nuclei and cell debris. The resultant supernatant was collected, and the protein concentration was determined using the Bradford assay (Bio-Rad Laboratories Inc., Hercules, CA, USA). Lysates containing equal protein contents (30 μg) were resuspended in Laemmli buffer and loaded on a 4–20% Mini-PROTEAN, TGX Stain-Free precast electrophoresis gel (Bio-Rad Laboratories, Inc., Hercules, CA, USA).

After electrophoresis, the proteins were transferred to a nitrocellulose membrane using a Mini Trans-Blot apparatus (Bio-Rad Laboratories Inc., Hercules, CA, USA), and the transfer of the protein onto the membrane was checked using a ChemiDoc molecular imager (Bio-Rad Laboratories, Inc., Hercules, CA, USA). For AQP8 and 9 immunoblotting, the nitrocellulose membrane was stripped using stripping buffer (Hi Media laboratories, Mumbai, India) and re-probed. The Western blot analysis of caspase-3 and Hsp70 was performed using the remaining protein lysates. 

Following gel transfer, the membrane was blocked with 5% non-fat milk diluted in TBS-T buffer (1.5 M NaCl, 200 mM TRIS-HCL, and 0.1% Tween 20, pH 7.2) at room temperature under constant motion for an hour.

After washes in TBS-T, the blot was incubated overnight at 4 °C with primary rabbit polyclonal antibodies directed against AQP7 (dil., 1:500, orb13253), AQP8 (dil., 1:500; orb101163), AQP9 (dil., 1:500; orb10127), caspase-3 (dil., 1:500), and Hsp70 (dil., 1:500; orb415817) (Biorbyt LLC, San Francisco, California, USA).

One day after incubation, the membrane was washed three times with TBS-T for 10 min and incubated with a Goat anti-Rabbit secondary antibody conjugated with horseradish peroxidase (HRP) (ImmunoReagents, Raleigh, NC, USA; dil., 1:1000) for 1 h at room temperature. After washing the membrane 3 times with TBS-T, ECL (Bio-Rad Laboratories, Inc., Hercules, CA, USA) was used to visualize the proteins, and an image was taken using the ChemiDoc molecular imager (Bio-Rad Laboratories, Inc., Hercules, CA, USA). For a positive control, canine testicular tissue extract was used, while for a negative control, the membrane filter was treated only with TBS-T used for antibody dilution but omitting the primary antibody. The standard molecular weight marker used was Precision Plus Protein™ All Blue Prestained Protein Standards (1–250 KdA, #1610373, Bio-Rad Laboratories, Inc., Hercules, CA, USA). Obtained bands for the examined proteins were visualized with Image Lab 6.0 program and were normalized to total proteins using the stain-free technology gels. The results are expressed as the intensity relative to that for normal tract.

### 2.4. Real-Time RT-PCR and Data Processing

Total RNA extraction, cDNA synthesis, RT-PCR, and sequencing were performed as previously described [38]. Briefly, total RNA was extracted using an Ultra-Turrax homogenizer from tissues in ice-cold TRIzol reagent (Life Technologies, Carlsbad, CA, USA). After chloroform extraction and isopropyl alcohol precipitation, the RNA was dissolved in RNAase-free diethyl dicarbonate (DEPC) water. Then, the RNA content was quantified using an Eppendorf BioPhotometer (Eppendorf AG, Basel, Switzerland). For the conventional and real-time RT-PCR reactions, primers specific for the selected canine transcripts were designed using Primer Express.

These specific primers, which amplify 200 (AQP7 and 8) and 150 (AQP9) base pairs, were designed based on the published GenBank gene sequences for the *Canis lupus familiaris* AQP7, 8, and 9 mRNAs.

The GenBank accession numbers and sequences of the primers are listed in Table 1. To prepare cDNA, 1 μg of total RNA was retrotranscribed using High Capacity cDNA Reverse Transcription Kits (Applied Biosystems, Carlsbad, CA, USA), according to the manufacturer’s instructions, using random hexamers as the primers. The PCR cycle conditions were as follows: 94 °C (30 s), 60 °C (30 s), and 72 °C (1 min) for 35 cycles and then 72 °C. The PCR products for canine AQP7, 8, and 9 were purified using a GFX PCR DNA and Gel Purification Kit (28-9034-70, GE Healthcare, Little Chalfont, Buckinghamshire, UK) and sequenced. Quantitative RT-PCR was used to study the mRNA transcript profiles for these genes. The real-time PCR reactions contained 1 μL of cDNA (20 ng/well) and 24 μL of SYBR Green Master Mix (Applied Biosystems, Carlsbad, CA, USA) containing specific primers. The PCR conditions were as follows: 50 °C for 2 min and 94 °C for 10 min, followed by 40 cycles of 94 °C for 15 s and 60 °C for 1 min. The GAPDH gene was also amplified in separate tubes under the same conditions to serve as an active endogenous reference for normalizing the quantification of the mRNA target.

Real-time detection was performed on an ABIPRISM 7300 Sequence Detection System (Applied Biosystem, Foster City, California, CA, USA), and data for the SYBR Green I PCR amplicons were assessed with the ABI 7300 System SDS Software. The relative expression for all the epididymal segments was quantified using the delta-delta Ct method (2^−ΔΔCt^), as described previously [52]. GAPDH expression was used for the normalization of the AQP7, 8, and 9 expression levels between the different samples.

### 2.5. Statistical Analysis

The results of the densitometric analysis and real-time RT-PCR are expressed as the mean ± standard deviation (SD) for all the values calculated. Variance analysis (ANOVA) for unpaired data and Tukey’s HDS test for independent samples were used to analyze the significance of differences in the relative abundances of the AQP7, 8, and 9 mRNA between the different epididymal segments in normal and cryptorchid dogs. In addition, the statistical significance of differences in the AQP7, 8, and 9 mRNA levels between the calibrator (normal epididymal segments) and cryptic counterparts were also determined using Student’s t-tests. All the experiments were performed in triplicate. A value of p < 0.05 was considered to indicate a statistically significant difference.

## 3. Results

### 3.1. Immunohistochemistry of AQP7, AQP8, and AQP9 in Normal and Cryptorchid Canine Epididymis

The results of the immunohistochemistry analyses of AQP7, 8, and 9 in the different epididymal segments of normal and cryptorchid dogs are shown in Figure 1, Figure 2 and Figure 3, respectively. Histologically, the normal epididymis is lined by pseudostratified epithelium consisting of three main regions: caput, corpus, and cauda. Principal cells represent the major cell type throughout the entire epididymis, followed by clear, narrow, apical, basal, and halo cells and the number, appearance, and function of these cell types vary in each segment of the epididymis. The height of the principal cells changes from tall columnar in the caput of the epididymis to low columnar in the corpus and cauda of the epididymal segments. The cryptorchidism results in reduced diameter of the duct and reduction in the length of the epidydimal epithelium. Fine histological alterations were observed between normal and cryptorchid epididymis, which consisted of the narrowing of the epididymal duct, flattering of the epithelial cells, disorganization of the tubular arrangement, and the presence of abundant interstitial tissue throughout the entire epididymis with the absence of spermatozoa in the lumen of the duct itself.

In the normal dogs, intensely stained AQP7-containing granules lined the entire cytoplasmic profile and the basal portion of the efferent ductules epithelium (Figure 1A, arrows). Positive material, weakly stained, was found in the apical (Figure 1B,D, arrowheads) and basal (Figure 1D, pounds) portions of the principal cells in the caput and cauda of the normal epididymis. Several basal cells positive for AQP7 were localized in the corpus (Figure 1C, double arrows). Some immunoreactive narrow cells were also detected in the caput of the normal epididymis (Figure 1B, line arrow), whereas in the cryptorchid epididymal segments, immunoreactive basal cells were observed in the caput and cauda epididymis (Figure 1E,F, double arrows). Dispersed positive granules immunoreactive to AQP7 were found in the cytoplasm of the principal cells of the cryptorchid cauda (Figure 1F), while the corpus did not exhibit any labeling more than the background (data not shown). For AQP7, in the normal epididymis, the density of immunoreactivity (IR) was higher in the cauda and corpus than the caput epididymis, and the labeling was confined to cytoplasmic domains.

In the normal subjects, AQP8 IR was observed in the cytoplasm of the efferent ductules epithelium (Figure 2A, arrows), particularly in the apical and basal portions. Immunoreactive material was also detected in the apical cytoplasm (Figure 2B–D, arrowheads) of the principal cells of all the epididymal segments. AQP8 IR was found in the basal portion (Figure 2C,D, pound) of the principal cells of the corpus and cauda epididymis. Some narrow cells reactive to AQP8 were intensely stained and localized in the caput epididymis (Figure 2B, line arrow). In the cryptorchid epididymal segments, however, basal cells (Figure 2E, double arrows) exhibited immunoreactivity in the caput. Immunopositivity was also observed in the apical portion of the principal cells of the cryptorchid cauda, with low intensity (Figure 2F, arrowheads). To better identify the positive cytotypes, inserts were added in Figure 1A,B and Figure 2D,E.

As described for the other peptides, AQP9 IR was found with the same immunolocalization in the efferent ductules (Figure 3A, arrows). A faint positivity was detected in the apical portion of the principal cells of the normal caput and cauda epididymis (Figure 3B,D, arrowheads). A few basal cells were positive (data not shown). In contrast to that for the other AQPs, intranuclear immunoreactivity was also observed for AQP9 throughout the segments of the normal epididymis (Figure 3B–D). However, there was a very faint immunoreactivity in the apical portion of the principal cells of the caput and cauda segments of the cryptorchid epididymis (Figure 3E,F, arrowheads).

Results of AQPs expression and their cellular distribution in the efferent ductules and epididymal segments of normal and cryptorchid dogs are summarized in the Table 2 and Table 3, respectively.

### 3.2. Analysis of Expression of AQP7, AQP8, and AQP9 in Normal and Cryptorchid Canine Epididymis by Western Blotting

The results of the Western blot analysis for AQP7, AQP8, and AQP9 are reported in Figure 4A.

For AQP7, immunostaining was clearly observable at 25–42 kDa and about 60 kDa for all the examined tracts of the epididymis of the normal and cryptorchid dogs, although differences in band intensity were apparent for both molecular weights between the normal and cryptorchid dogs. In particular, the data from the densitometric analysis (Figure 4B) demonstrated similar band intensities for AQP7 in all the tracts of the epididymis for the normal dogs, while a slight increase was observed only for the caput epididymis in the cryptorchid dogs relative to that in the normal dogs. A different pattern of AQP7 expression was observed in the corpus and cauda regions of the epididymis of the cryptorchid dogs, where the levels of the protein were relatively low.

The Western blotting image in Figure 4A shows, for AQP8, two bands corresponding to 35–48 kDa and ~60 kDa for all the tracts, although differences in intensity were particularly evident for the tracts of the cryptorchid dogs. The relative intensity of AQP8 (Figure 4B) was similar for all the epididymal tracts of the normal dogs. By contrast, while moderately high levels were observed for the caput and corpus of the epididymis of the cryptorchid dogs relative to those for the normal dogs, a lower intensity was observed for the cauda region of the epididymis.

The Western blotting image shows (Figure 4A), for AQP9, an intense band corresponding to ~60 kDa and two weak bands corresponding to 35–48 kDa for all the tracts of the normal and cryptorchid epididymis, albeit slight differences are observed between different tracts and normal vs. cryptorchid conditions. Densitometric analysis (Figure 4B) reveals a lower intensity for the corpus and cauda tracts in the epididymis of the cryptorchid dogs than those of the normal dogs, while only a slight increase in intensity is observed for the caput epidydimal tract of the cryptorchid dogs.

### 3.3. mRNA Expression of AQP7, AQP8, and AQP9 in Normal and Cryptorchid Canine Epididymis According to Real-Time RT-PCR

Real-time RT-PCR analysis was performed to determine the expression of these AQPs in different segments of the epididymis from the normal and cryptorchid dogs. As shown in Figure 5A, in the normal dogs, all three AQP mRNAs were expressed in all the segments of the normal epididymis, with similar expression levels except for AQP7, which showed the highest value in the cauda. The observations were different for the cryptorchid dogs; AQP7 and AQP8 showed low mRNA levels in the corpus and cauda and high levels in the caput of the epididymis (Figure 5B); by contrast, AQP9 mRNA was high in all three segments of the epididymis, with a slight increase in expression in the caput (Figure 5B). In addition, this analysis showed a similar trend for the AQP7 and AQP8 transcripts in the segments of the epididymis of the cryptorchid dogs as compared with the normal dog samples. Increases in the mRNAs were observed for the caput and corpus, while a decrease was shown in the cauda (Figure 5C). The increase in the AQP7 and AQP8 mRNAs relative to the normal sample levels was particularly high only in the caput. However, AQP9 mRNA was slightly less abundant in all the epididymal segments of the cryptorchid dogs compared to in the normal dogs (Figure 5C).

### 3.4. Western Blot Analysis of Hsp70 and Caspase-3 Expression in Normal and Cryptorchid Canine Epididymis

To better investigate the possible role of AQPs in the epididymis and their relationships with factors such as heat and oxidative stress, which are both involved in cryptorchidism, Hsp70 and caspase-3 were evaluated (Figure 6A). The densitometry showed an increase in Hsp70 protein expression in the cryptorchid condition compared to the normal dog samples (control) in all the segments of the epididymis, with the highest and most significant value in the caput followed by the corpus and cauda regions (Figure 6B).

Caspase-3 expression showed a reduced expression in both the epidydimal segments of the caput and corpus of cryptorchid dogs compared to the control, with a significant decrease for the first epididymal segment. In the cauda, the protein showed its highest expression level with a value of relative intensity > 2.5 compared to the control. (Figure 6B).

## 4. Discussion

In the present study, we examined the localization and expression of the AQP7, 8, and 9 proteins and their mRNAs in the efferent ductules and epididymis of normal and cryptorchid dogs. To clarify the possible roles played by the examined AQPs in the epididymal tissue, we analyzed the expression of Hsp70 and caspase-3, which are well known to be involved, among numerous other factors, in the heat stress and oxidative mechanisms related to cryptorchidism [53,54,55].

Our results obtained by immunohistochemistry demonstrate the presence of the examined AQPs in different epithelial cells of the efferent ductules and of all the tracts of the epididymis in both normal and cryptorchid dogs, although a less dense distribution of cellular immunoreactivity to AQPs was shown in the cryptorchid dogs than in the normal dogs. Moreover, the analyses of the localization and relative protein and mRNA expression of the AQPs showed significant differences among the examined segments of the epididymis.

In normal subjects, the results of AQP8 and AQP9 IR demonstrated their presence in the basal and apical portions of the cells of the efferent ductules epithelium. Similar cellular distribution for AQP9 was well described by Domeniconi et al. (2007) [39], who reported its expression in the apical brush border of non-ciliated cells. The possible role played by AQP9 could be to allow the movement of water across the epithelia to improve the passage of glycerol, considering that this molecule serves as a metabolic substrate for sperm to produce CO_2_ [44,56].

Conversely, our results relative to the positivity of epithelial cells to AQP7 disagree with that of Domeniconi et al. (2008) [44], leaving us to attribute this discrepancy to different factors, including a different sensitivity of the used antibody and the methods of tissue preparation and immunocytochemical procedures utilized in each case. Of note, the only immunohistochemical study in rats showed both AQP7 and AQP9 in the efferent ductules, with a moderate expression of AQP7 either over the microvilli of epithelial cells or along the basolateral plasma membranes [26,44]. The co-expression of different AQPs along the epithelial cells of this reproductive tract of dogs suggests their possible role in regulating luminal fluid composition to promote spermiogenesis [57]. Moreover, considering that AQPs are influenced by hormones, the absence of AQP IR demonstrated in the cryptorchid dogs could be due to a reduced production of androgens because of the cryptorchid condition, as previously observed in rats [58]. Additional experimental data using mice deficient in α-estrogen receptors (αESRKO mice) demonstrated a decreased AQP1 and AQP9 expression in the efferent ducts, confirming the important hormonal control on the efferent ductules function in retention of water [59].

In normal epididymal segments, our results demonstrated slight differences in the cellular distributions of the aquaglyceroporins AQP7 and AQP9 along the epididymal segments; AQP7 was localized in the cytoplasm of the principal cells of the caput and cauda segments and basal cells only in the corpus, whereas AQP9 was distributed only in the principal cells of the entire epididymal tract, albeit the immunoreactivity was exclusively intranuclear. The AQP7 expression was also confirmed by Western blot demonstrating bands at ~30 kDa (faint band) and ~42–60 kDa, respectively. This result confirms previous reports on dogs [44] showing only the first band, while protein extracts from bull spermatozoa only displayed a band corresponding to 45 kDa [60]. The presence of AQP7 in the epididymal basal cells was interpreted as being part of the intricate cooperative participation in the removal of water and/or small uncharged molecules from the epididymal lumen throughout the epididymis [44]. The basal cells are numerous in the corpus as previously demonstrated [61,62,63], and their particular morphological characteristic gives them huge potential for the movement of water, small uncharged molecules, or both in the epididymis morphology [26].

The distribution of AQP9 in all the segments of the normal canine epididymis as well as its expression as a band corresponding to 30 kDa confirms previous data by Domeniconi et al. (2007) [39]. Other previous findings show this protein in the epididymis of rats [64] and mice, respectively [65], albeit with a localization in the apical stereocilia of principal cells. AQP9 could participate in the trafficking of water and/or solute permeability of these epithelia, contributing to the processes taking place in this tissue. Furthermore, the intranuclear localization of AQP9 observed in the epithelial cells of the epididymis in our study has also been reported in sheep and post-pubertal pigs [37], which suggests a putative role in purine and pyrimidine transport [66].

The co-expression of AQP7 and AQP9, having in common the ability to allow the cellular membrane trafficking of water, small solutes, and glycerol, albeit distributed in different epididymal cells, could sustain the effective transport of glycerol and glycerylphosphorylcholine from the epithelium to the lumen, where these molecules play a pivotal role in the process of sperm maturation [67].

AQP8 IR was shown in the apical portion of the principal, basal, and narrow cells of the normal epididymal segments in a region-specific manner. The presence of AQP8 was also observed, although with slight immunoreactivity, in the rat epididymal basal cells [68], and its role could be associated with that assumed for these cells at the level of the conductive airways. At this level, AQP8 could be involved in the differentiation of epithelial cells during maturation or after recovery from injury to the airways [68]. Notoriously, basal cells present in all mammalian species are supposed to sustain a stem function in the renewal of the epithelium [69,70,71]. The presence of these AQPs at the level of the epididymis, as well as the suggestion of their involvement in different processes, are all coordinated by hormonal control that could include: (1) the water absorption from the testicular fluid; (2) the creation of a luminal microenvironment in physical and nutrient parameters essential to spermatozoa growth; and (3) the transfer of solutes and glycerol to sustain spermatozoa along the cauda epidydimal segment.

In the cryptorchid condition, significant and intense differences were observed in terms of AQP localization and expression in all the segments of the epididymis. AQP7 and AQP8 showed a moderate shift in their cellular localization compared to control passing from principal, narrow, and basal cells (control) to the principal and basal cells, albeit with a faint IR. AQP9 IR showed a cellular localization along the epididymal segments similar to that observed for the control. The presence of AQP7 and AQP8 in the basal cells in the course of this disease could hypothesize a different and new role exerted by these proteins. Basal cells have been largely investigated for their multifaceted roles, facilitating the cell–cell crosstalk [72] and functioning as luminal sensors of principal cells [73] throughout the prostaglandin production [74]. Particularly noteworthy is the participation of basal cells in immune functions and their possession of genes that encode proteins involved in cell adhesion, cytoskeletal arrangement, ion transport, cellular signaling, and inflammatory responses [75]. Moreover, Pinel et al. (2019) [76] have reported a new and intriguing involvement of these cells as responsible for the equilibrium between apoptosis and cell renewal in the epididymal epithelium. The role of these cells and the associated AQP expression could be associated with the pathway of Hsp70 and caspase expressions. In particular, in the caput where AQP IR was quite exclusively associated with these cells (only a faint AQP9 IR is associated with epididymal principal cells) and a high level of Hsp70 expression was shown, a mechanism of homeostasis regulation at the level of the lumen microenvironment to counteract the oxidative stress responsible for sperm cell death could be hypothesized.

This hypothesis is corroborated by evidence that the cellular localization of Hsp70 is selective for basal cells in the human epididymal epithelium [53] and the fact that we demonstrated, by Western blotting, high AQP7 expression in the caput of the cryptorchid canine epididymis. In addition, Lu et al. (2013) [77] proposed that other mechanisms, such as necrosis or inflammation, could be preferred over apoptosis at this level. Prieto-Martinez et al. (2017) [78] evaluated AQP expression in boar sperm in relation to sperm cryotolerance and demonstrated that AQP7 was expressed in ejaculates exhibiting good freezability (GFE), suggesting its possible role as a freezability marker.

Similarly, for AQP9, the higher expression observed in the caput of the cryptorchid dogs, associated with high levels of Hsp70, could suggest its involvement in the protection from stress. Of note, previous in vitro studies demonstrated an upregulation of AQP9 in rat intestinal epithelial exposed to hypertonic stress [79].

The localization of AQPs in the cauda region of the epididymis either along the basal or principal cells could suggest different roles of these proteins in this epididymal segment. In particular, the principal cells where AQPs are localized could be involved in the wave of apoptosis determined by androgen loss. AQP9 could play a role in this mechanism since its downregulation in mouse Sertoli cells exposed to excess of 17β-Estradiol (E2) has been demonstrated [80]. Similarly, AQP9 downregulation has also been shown to be associated with apoptosis and caspase-3 increase in retinal ganglion cells (RGCs) exposed to stress, thus demonstrating the role of this protein in transporting lactate as an energy substrate and as a source of ROS scavengers. Indirectly, these findings suggest that the cryptorchid condition could influence AQP9 preventing its role in transporting glycerol in this district and thus could induce cell death [81].

Thus, we could hypothesize that cryptorchidism in dogs could impair a general dysregulation of the epithelial cells along the efferent ductules and epididymal segments, influencing the expression of AQPs. The particular AQP cellular distribution and the modulation of Hsp70 and caspase-3 (Figure 7) could suggest their different responses to cellular alterations are probably associated with mechanisms ranging from inflammation or necrosis to male germ cell death, which are often observed in the cryptorchid condition or in the experimental condition of the efferent duct ligation [82,83]. Further studies are needed to characterize possible segment-specific regulation of epididymal cell apoptosis as well as other possible mechanisms involved in the canine cryptorchid condition.

## 5. Conclusions

The different patterns of the expression and distribution of AQPs in the efferent ductules and in the epididymal segments of cryptorchid dogs suggest their involvement in luminal microenvironment modifications that are peculiar characteristics of this pathophysiological condition. The cryptorchid condition characterized by multiple associated processes (androgen decrease, heat stress, etc.) could influence the sensitivity of AQPs at the cellular level, modulating their activity. Further investigations using in vitro models are needed to clarify the exact role played by each of the examined AQPs, and a simultaneous examination of the different immunological, oxidative, and inflammatory states of the epididymis would improve the knowledge on the pathogenesis of canine cryptorchidism.

## Figures and Tables

**Figure 1 animals-11-01539-f001:**
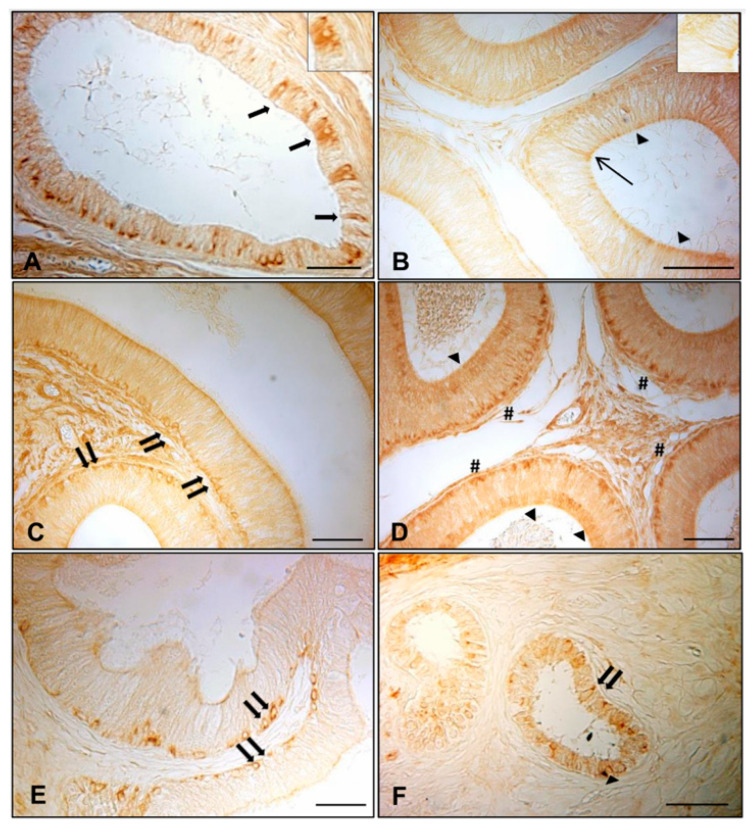
Immunohistochemical staining of AQP7 in normal and cryptorchid efferent ductules and epidydimal segments. In the normal dogs, AQP7 was distributed in (**A**) the epithelium lining the efferent ductules (arrows), (**B**,**D**) the apical portion of the principal cells of caput and cauda (arrowheads), (**D**) the basal portion of the cauda (pounds), and (**B**) in some narrow cells of the caput (line arrow). (**C**,**E**,**F**) Positive basal cells were observed in the corpus of the normal epididymis and in the caput and cauda of the cryptorchid epididymis (double arrows). Bar: 25 micron.

**Figure 2 animals-11-01539-f002:**
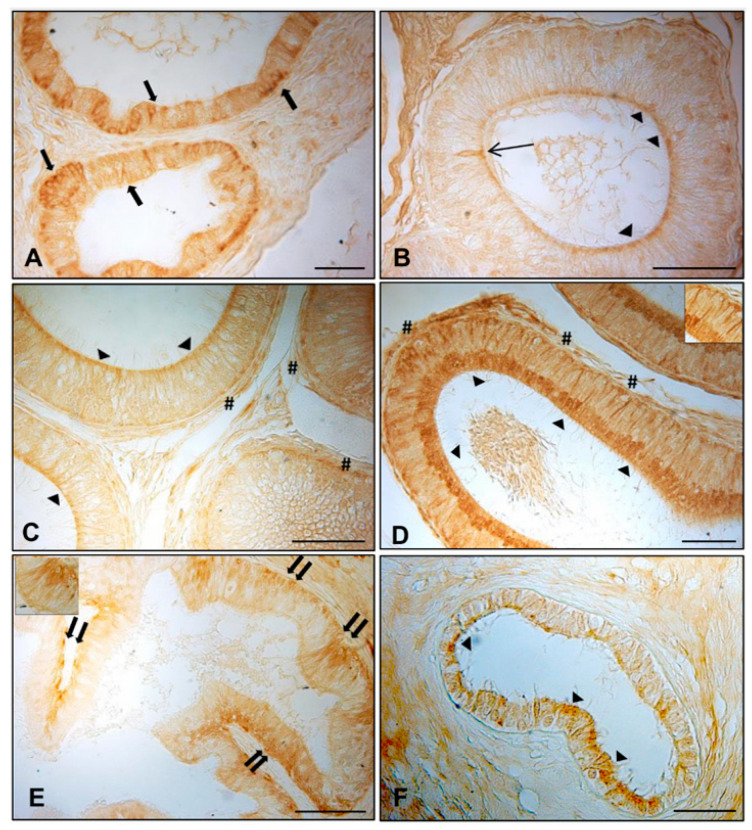
Immunohistochemical staining of AQP8 in normal and cryptic efferent ductules and epididymal segments. In the normal dogs, AQP8 IR was described in (**A**) cytoplasm of the efferent ductules epithelium (arrows), (**B**–**D**) the apical cytoplasm (arrowheads) of the principal cells of all the epididymal segments, (**C**,**D**) the basal portion of the principal cells of the corpus and cauda (pounds), and (**B**) in some narrow cells of the caput (line arrow). In the cryptorchid epididymis, basal cells (double arrows) were observed in the caput (**E**), while only the apical portion of the principal cells of the cauda (arrowheads) exhibited immunoreactivity (**F**). Bar: 25 micron.

**Figure 3 animals-11-01539-f003:**
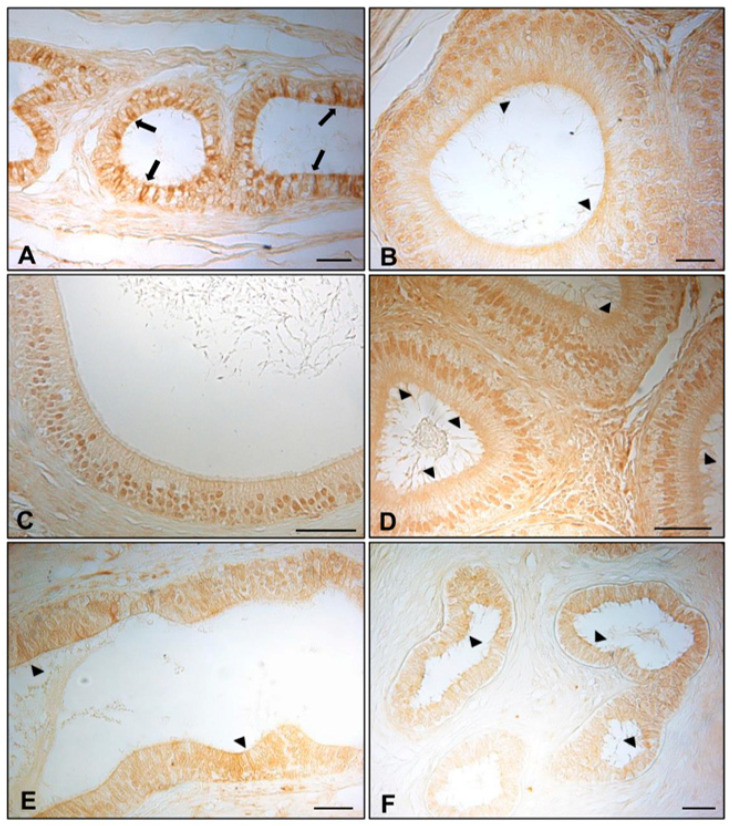
Immunohistochemical staining of AQP9 in normal and cryptic efferent ductules and epididymal segments. In the normal dogs, AQP9 IR was described in (**A**) cytoplasm of the efferent ductules epithelium (arrows) and (**B**,**D**) the apical cytoplasm (arrowheads) of the caput and cauda epididymis. Intranuclear IR was also observed for AQP9 in all epididymal segments (**B**–**D**). (**E**,**F**) A very faint immunoreactivity was detected in the apical portion of the principal cells in the caput and cauda segments of the cryptorchid epididymis (arrowheads). Bar: 25 micron.

**Figure 4 animals-11-01539-f004:**
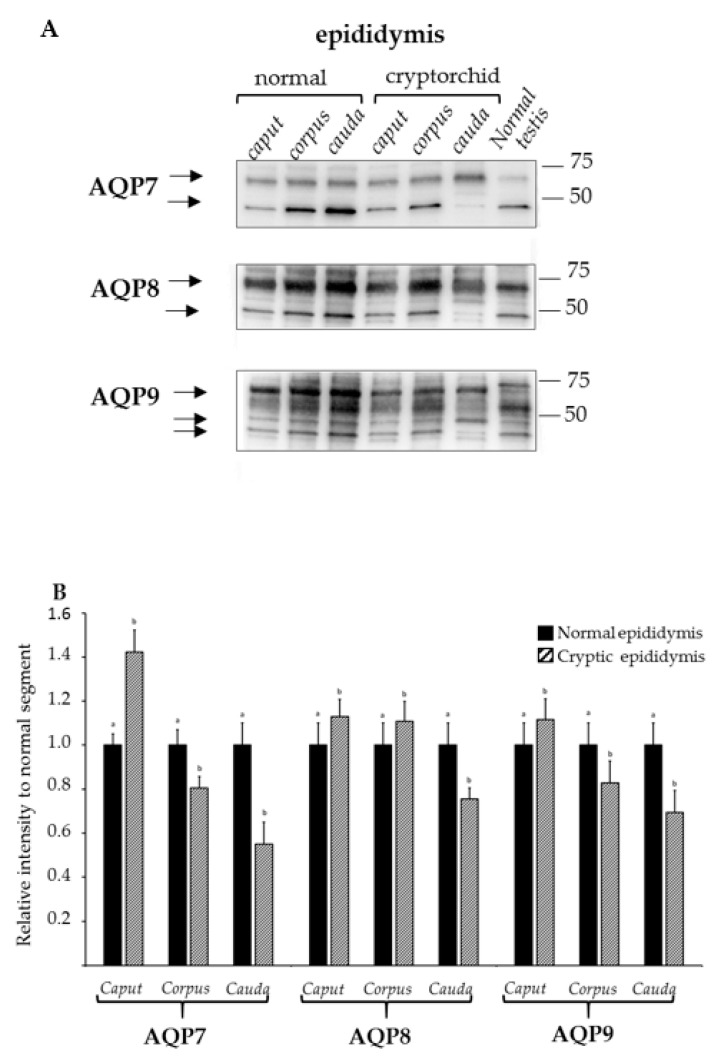
Western blotting of AQP7, AQP8, and AQP9 protein expression in the epididymis regions (caput, corpus, cauda) of normal and cryptorchid dogs. (**A**) Representative immunoblot; (**B**) densitometric analysis of AQP7, AQP8, and AQP9 protein expression. Results presented are the mean ± standard deviation of three independent experiments. Different letters depict differences between the examined groups (*p* < 0.05).

**Figure 5 animals-11-01539-f005:**
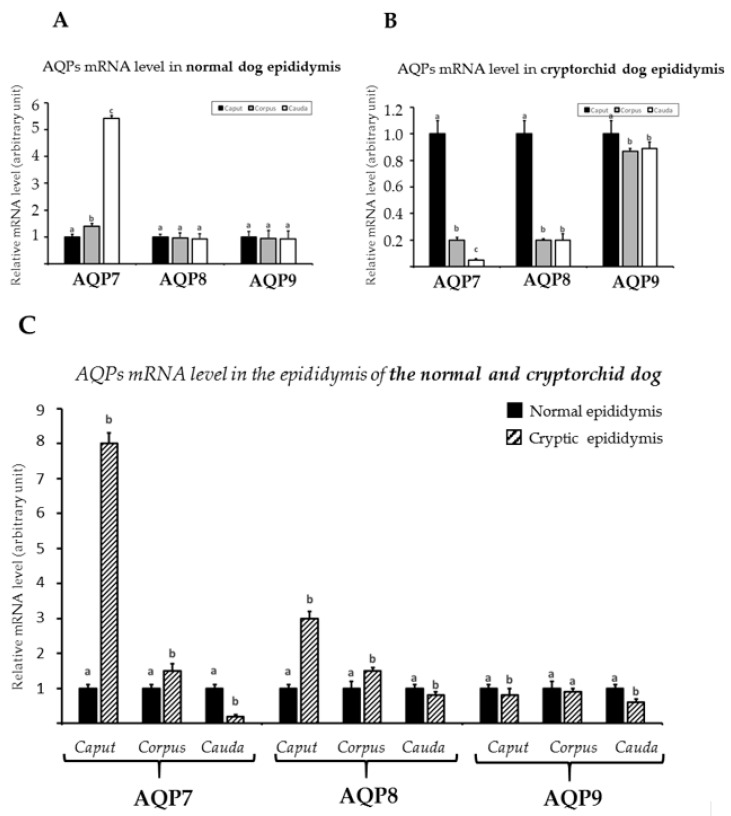
AQP7, AQP8, and AQP9 mRNA level expression in the normal (**A**,**C**) and cryptorchid dogs (**B**,**C**). (**A**) The calibrator is the normal epididymis caput. (**B**) The calibrator is the cryptic epididymis caput. (**C**) The calibrator is the normal epididymal segment. Each plotted value corresponds to the mean ± standard deviation (SD) obtained from three independent experiments. Different letters depict differences between the examined groups (*p* < 0.05).

**Figure 6 animals-11-01539-f006:**
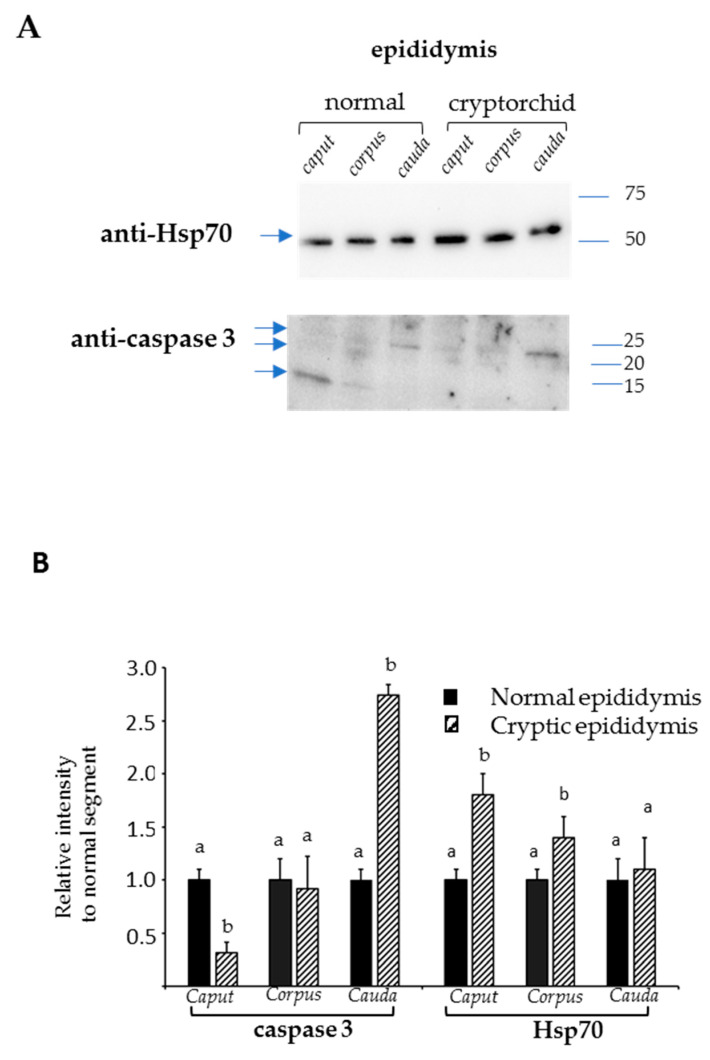
Western blot analysis of Hsp70 and caspase-3 in the epididymal tracts of normal and cryptorchid dogs (**A**). Densitometric analysis of the expression levels of Hsp70 and caspase-3 (**B**). Different letters depict differences between the examined groups (*p* < 0.05).

**Figure 7 animals-11-01539-f007:**
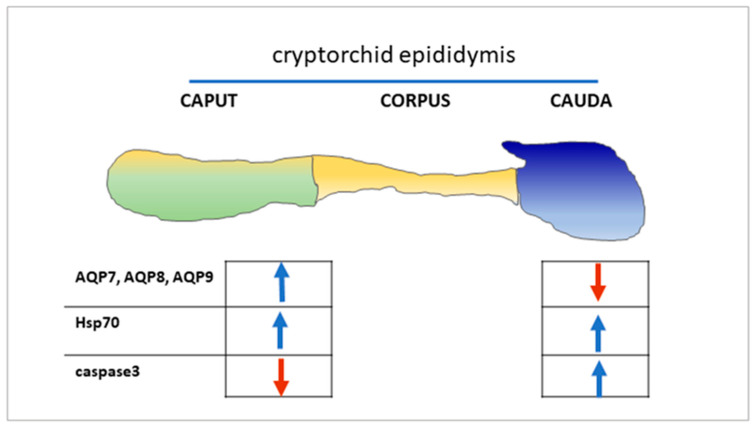
Regulation in the expression levels of AQPs, Hsp70, and caspase-3 in the epididymal tracts of cryptorchid dogs.

**Table 1 animals-11-01539-t001:** Primers for real-time RT-PCR.

Gene Target	Sequence 5′–3′	Genbank Accession Number
***AQP7***		XM_531973.5
Forward	TGTCCCCACCCCCAATG	
Reverse	TCGTATATTTGGCAGCTTTCTC	
***AQP8***		XM_014114532.1
Forward	AAACATCAGTGGAGGACATTTCAA	
Reverse	GCTCCTGGACTGTCACAAAGG	
***AQP9***		XM_544701.4
Forward	CCTTCCCTGCGAATCACAA	
Reverse	AGGTGCATCGCTTGATGTAGAG	

**Table 2 animals-11-01539-t002:** Distribution of AQP7, 8, and 9 IRs in the efferent ductules and epididymis of normal and cryptorchid dogs.

NORMAL		AQP7	AQP8	AQP9
Epididymis	Efferent ductules	+++	+++	+++
	Caput	++	++	++
	Corpus	++	++	++
	Cauda	+++	+++	++
**CRYPTORCHID**				
Epididymis	Efferent ductules	-	-	-
	Caput	++	+++	++
	Corpus	-	-	-
	Cauda	+++	+++	++

+++: high density; ++: moderate density; +: low/mild density; - absence.

**Table 3 animals-11-01539-t003:** Cellular distribution of AQP7, 8, and 9 in the epididymis of normal and cryptorchid dogs.

			AQP7				AQP8				AQP9	
	Epididymal Segment			Cell Types								
		P	B	N		P	B	N		P	B	N
**NORMAL**	Caput	+	-	+		+	-	+		+ *	-	-
	Corpus	-	+	-		+	-	-		+ *	-	-
	Cauda	+	-	-		+	-	-		+ *	+/-	-
	Caput	-	+	-		-	+	-		+	-	-
**CRYPTORCHID**	Corpus	-	-	-		-	-	-		-	-	-
	Cauda	+	+	-		+	+	-		+	-	-

P = principal cells; B = basal cells; N = narrow cells; + presence; - absence; +/- rare; + * intranuclear presence.

## Data Availability

Not applicable.

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
