# Peer review of "Aquaporins Are Differentially Regulated in Canine Cryptorchid Efferent Ductules and Epididymis"

_animals, 2021, doi:10.3390/ani11061539_

Round 1

Reviewer 1 Report

The authors modified the manuscript according to the comments from the referee. The changes were positive and relevant to the manuscript which describes the distribution and expression of AQP7, AQP8, and AQP9 in the efferent ductules and epididymal regions from normal and cryptorchid dogs. The present paper is appropriate for this journal and adds a new perspective about the expression and role of AQPs in the dog male reproductive system.

Author Response

We thank you for the positive answer

Reviewer 2 Report

In the present study, the aim was to assess the localization and expression patterns of AQP7, AQP8, and AQP9 in the efferent ductules and different regions of the epididymis in normal and cryptorchid dogs. Although the manuscript is written well and the study design was comprehensive but I recommend that the manuscript has to be corrected before its acceptance for publication.

1. Fig. 1, 2, and 3

    - It would be helpful if the authors provide the inset with a magnified image in each figure.

2. Fig. 4

    - The authors are asked to provide how to normalize the WB data and need to explain AQP9 expression in the result section.

    - In addition, the authors are asked to use common loading control proteins such as bet-actin for quantification or provide better quantification methods to be confirmed.

3. Fig. 6 

    - The authors should provide IHC images performed with Hsp70 and caspase 3 antibodies.

4. A detailed description of the histological differences between normal and cryptorchid epididymides is lacking. Please provide it.

5. Lines 574-579. I cannot understand how the authors conclude that the alterations of Hsp70 and caspase 3 by AQPs changes correlate with male germ cell death in the epididymis. Please explain this.

6. The overall manuscript is very difficult to read and bringing the pieces together was confusing. I believe the manuscript must be revised with the clear results.

Minor:

Line 285. Change to Figure 1F

Author Response

We thank you the referee for the suggestions. Our revision letter is here alleged

Round 2

Reviewer 2 Report

I appreciate to authors' efforts to modify the manuscript.

This manuscript is a resubmission of an earlier submission. The following is a list of the peer review reports and author responses from that submission.

Round 1

Reviewer 1 Report

In present investigation the authors proposed examined region-specific distribution pattern of the AQPs 7, 8 and 9 of normal and cryptorchid dogs. These analyze initially direct knowledge of the reproductive aspects in dogs and add new information about cryptorchidism. Despite this, there are concerns which the authors must address:

The introduction should provide more information about Hsp70 in epididymis subjected to heat stress conditions.

The presentation of the results related to immunohistochemistry should be improved. The quality of the images makes it difficult to understand the location of AQPs along the epididymal epithelium.

In general, the discussion would benefit from a more detailed insight into the contribution of AQPs proteins to dog reproductive biology. The discussion could better explore the relation among AQPs, basal cells and HSP70. In this sense, the conclusion should also be revised.

Specific comments:

line 247: the authors described: “Several basal cells positive for AQP7 were localized in the corpus”.  Authors could add the information that these basal cells positive for AQP7 were also localized in the cauda.

line 254:  Based on the figures presented, it is not possible to say that there is a difference in the intensity of the immunoreactivity (“For AQP7, the intensity of immunoreactivity was higher in  the cauda and corpus than the caput epididymis, and the labeling was confined to cytoplasmic domains”).

line 258:  Apparently there was a misunderstanding in this sentence since the epithelium of the efferent duct is not considered an epididymal segment.

Figures 1, 2 and 3 were presented with low quality and do not allow to accurately describe the location of AQPs. In addition, AQP labeling does not appear to be specific. Thus, the authors should provide pictures with higher quality. It seems that the time used for DAB treatment was longer than recommended. In order to improve the technique, the sections should have been counterstained with hematoxylin.

The Figure Captions should be revised. In Figure Caption 1: the letter C is missing; In Figure Caption 2: it is not clear which region is indicated in the letter E.

line 388: I suggest that the caspase results be rewritten.

line 432:  this sentence should be reconsidered, since AQP9 is present in throughout the entire epididymis.

line 440: Previous data reported in the rat and mouse epididymis described AQP9 in the apical stereocilia from principal cells. Therefore, these results are different from the data presented in the present study. This sentence needs to be revised.

line 448: I suggest that the AQP8 discussion be rewritten considering the basal cells. The authors cited (55) suggest the possible involvement of AQP8 in the “differentiation from basal cells to more differentiated cell types in these epithelia”

line 517: This sentence should be reconsidered, since there was no dowregulation of Hsp70.

Author Response

Dear Editor,

we appreciated the evaluation of the reviewers #1 and #2 in respect to our manuscript entitled “Aquaporins Are Differentially Regulated in Canine Cryptorchid Epididymis”.

In this regard, Reviewer’s comments have allowed us to significantly improve the manuscript: a) specifying some details on AQPs immunohistochemistry data, b) adding negative control in western blotting; c) adding more detailed insight into the contribution of AQPs to the dog reproductive biology, d) modifying the discussion to comment exclusively the results obtained.

Please find below a point-by-point reply to the Reviewers’ comments (reported in bold).

As requested, we used for the revision the file submitted first time and we applied the mode of “Track Changes”. Major changes are highlighted in yellow within the main body of the text.

We hope that the current revision could be suitable for the publication in Animals.

Regards,

Caterina Squillacioti, Nicola Mirabella, Giovanna Liguori, Giuseppe Germano and Alessandra Pelagalli

Author's Notes to Reviewer 1

We wish to thank you for the time spent on our manuscript (Paper Animals -1141423) entitled “Aquaporins Are Differentially Regulated in Canine Cryptorchid Epididymis” and for the opportunity to submit a revised version of the manuscript modified in response to your comments.

My co-authors and I modified several sentences in the manuscript accordingly. Below you will find a point-by-point answer to all remarks and changes we made in the manuscript. The changes in the sentences of the manuscript were made by ticking the original text and by adding the new text. Every change was highlighted in yellow.

In Bold are reported the Reviewers comments.

Comments of the Reviewer #1:

GENERAL Comments and Suggestions for Authors

In present investigation the authors proposed examined region-specific distribution pattern of the AQPs 7, 8 and 9 of normal and cryptorchid dogs. These analyze initially direct knowledge of the reproductive aspects in dogs and add new information about cryptorchidism. Despite this, there are concerns which the authors must address:

  1. The introduction should provide more information about Hsp70 in epididymis subjected to heat stress conditions.
  2. The presentation of the results related to immunohistochemistry should be improved. The quality of the images makes it difficult to understand the location of AQPs along the epididymal epithelium.
  3. In general, the discussion would benefit from a more detailed insight into the contribution of AQPs proteins to dog reproductive biology. The discussion could better explore the relation among AQPs, basal cells and HSP70. In this sense, the conclusion should also be revised.

General

We thank the Reviewer for his/her very careful reading of our manuscript and for his/her important comments.

Answer to point A: As suggested by the reviewer, we added some sentences describing the role of Hsp70 in the epididymis subjected to heat stress (introduction, page 2 lines 87-94).

Answer to point B: We regret very much for the presented low quality of the immunohistochemistry images. We hypothesize that during the transfer of images into the text word file, they could have lost their high quality that we can see using normal viewer program. To avoid this inconvenience, we have added images (high quality) as separate files in supplementary materials.

Answer to point C: We modified the discussion by adding new insight on the role of AQPs in the reproductive biology of dog (discussion, page 15, lines 518-523) and some additional information on the possible role of AQPs in relation to HSP70 and basal cells (discussion, page 15, lines 536-544). Moreover, as suggested, we modified conclusions (lines 584-586). 

Specific comments:

line 247: the authors described: “Several basal cells positive for AQP7 were localized in the corpus”. Authors could add the information that these basal cells positive for AQP7 were also localized in the cauda.

answer: We thank the reviewer for his/her/punctual observation but in the normal subjects we have found basal cells AQP7 IR only in the corpus because in the cauda we described as positive the basal portion of the principal cells. We added these information (results, page 6, lines 277-280).

line 254:Based on the figures presented, it is not possible to say that there is a difference in the intensity of the immunoreactivity (“For AQP7, the intensity of immunoreactivity was higher in the cauda and corpus than the caput epididymis, and the labeling was confined to cytoplasmic domains”).

answer: We appreciate the suggestion of the reviewer and in order to better describe the expression of AQPs along efferent ducts and regions of canine epididymis in both normal and cryptorchid condition, we summarized their distribution and their different positive intensity in Table 1 and 2.

line 258: Apparently there was a misunderstanding in this sentence since the epithelium of the efferent duct is not considered an epididymal segment.

answer: As rightly indicated by the reviewer, we provided to differentiate the efferent ductules from the epididymal segments in the entire manuscript. In addition, we added several sentences in the discussion to analyze AQPs function in this tissue (discussion, page 14, lines 460-482).

Figures 1, 2 and 3 were presented with low quality and do not allow to accurately describe the location of AQPs. In addition, AQP labeling does not appear to be specific. Thus, the authors should provide pictures with higher quality. It seems that the time used for DAB treatment was longer than recommended. In order to improve the technique, the sections should have been counterstained with hematoxylin.

answer: As previous reported in the answer to general comments, in the reviewing process of the manuscript, we have submitted immunohistochemistry images of high quality as separate files in supplementary material. The time used for DAB treatment was from two to five minutes, depending on the primary antibody. The duration of DAB incubation was determined through pilot experiments and was then held constant for all of the slides. In addition, thank you very much for your suggestion regarding the counterstaining of the slides, but we cannot counterstain our slides because AQP9 localizes also in the nuclei and we think that this could damage and/or falsify DAB signal under the microscope, i.e. it could be added to that of the DAB. 

The Figure Captions should be revised. In Figure Caption 1: the letter C is missing; In Figure Caption 2: it is not clear which region is indicated in the letter E.

answer: As suggested by the reviewer that we thank for his/her punctual attention, we modified by adding letter C in Figure 1 caption and specifying the region indicated with letter E

line 388: I suggest that the caspase results be rewritten.

answer: As suggested by the reviewer, we rewrote the results referring to caspase-3 (page 13, lines 435-438)

line 432: this sentence should be reconsidered, since AQP9 is present in throughout the entire epididymis.

answer: As indicated by the reviewer, we modified the sentence specifying that AQP9 was distributed along the principal cells of the entire epidydimal epididymis albeit with a nuclear presence (discussion, page 14, lines 486-487)

line 440: Previous data reported in the rat and mouse epididymis described AQP9 in the apical stereocilia from principal cells. Therefore, these results are different from the data presented in the present study. This sentence needs to be revised.

answer: We agree perfectly with the reviewer and regret for this inaccuracy of the description of AQP9 expression in the rat and mouse epididymis. As suggested, we modified the sentence (discussion, page 15, lines 500-501).

line 448: I suggest that the AQP8 discussion be rewritten considering the basal cells. The authors cited (55) suggest the possible involvement of AQP8 in the “differentiation from basal cells to more differentiated cell types in these epithelia”

answer: Accordingly, to the suggestion of the reviewer that we thank for helping us to improve the quality of the discussion, we modified the discussion of the AQP8 introducing the possible role in the basal cells (discussion, page 15, lines 513-518).

line 517: This sentence should be reconsidered, since there was no dowregulation of Hsp70.

answer: We perfectly agree with the reviewer regret for our inattention. We modified this part of discussion completely and modified the sentence describing possible mechanisms related to the regulation of Hsp70 (discussion, page 16, lines 568-573).

Reviewer 2 Report

Title: Aquaporins Are Differentially Regulated in Canine Cryptorchid Epididymis

This manuscript described the expression patterns of specific aquaporins (7, 8 and 9) in the canine cryptorchid and normal epididymis using real time RT-PCR, immunohistochemistry (IHC) and western blot (WB) techniques.  No other publications have looked at aquaporins in these tissues in the canine cryptorchid model.  Thus, this provides some potentially interesting information and can be used in comparisons across species.  The results showed different patterns for the distribution and expression of the examined AQPs, upregulated in the caput and downregulated in the cauda region of the canine cryptorchid epididymis. These findings were associated together with the HSP70 and caspase-3 expressions to shed some light about the roles of these water channels in this pathophysiological condition. 
I feel that the manuscript was well written, and that the methodology was well explained. While the discussion was relatively long, I feel that was essential to explain the results found, especially considering the number of proteins (AQP-7, -8, -9, HSP70 and caspase 3) that this study was investigating.
However, there are several concerns, both major and minor regarding the manuscript.

The major concerns are:
1) The IHC photographs presented in the reviewers file were of low quality, so it was difficult to discern what it was described. 
2) The IHC controls (positive and negative) are described in the Materials and Methods (lines 149-151), but the results of the negative controls should be included in the plates to avoid misinterpretation of background or unspecific immunolabeling (e.g. AQP-9 and nuclear labeling). In addition, the results/description of the IHC (specific cell location and /or tissue) for the AQP-7, -8 and -9 should be summarized in a table format as many of the papers cited by the authors including approximately quantification of stain in intensity and/or number of cells (%).
3) There is no mention of negative controls described in the WB analysis, consequently neither in the results. Although the positive control is shown in the WB image, negative controls are necessary when the antibodies are not generated against the proteins of the specific species, in this case canine. 
4) A morphological description of the lesions observed in the cryptorchid epididymis (epithelial lining and diameters of the ducts) would be appreciated (comparing D and F of the plates cauda epididymal segments).

Minor concerns:
1) The manuscript needs some minor language review.
2) The use of “efferent duct” seems incorrect. There are several efferent “ductules” (ductuli efferentes) from the capital pole of the rete testis to the single tube forming the caput of the ductus epididymis. With that said, the size of the efferent ductules are considerably smaller in diameter than the caput epididymis and this is not perceived in the IHC photos (A versus B in figures 1, 2 and 3).
3) The manuscript needs some editorial review:
Line 280: “B-D” instead of “B,D” or else describe C in the text.
Line281: the symbol used in photograph D is not an asterisk (*) but a # (pound or number symbol).
Line 293: same as above.
Line 390: Figure 5B should be Figure 6B. Also, the reviewer is confused about the comparison in the graph displayed in line 394 (Figure 6B) and the description presented in 389-390. Are the comparisons and the significant differences in the figure from normal caput to cryptic caput? Normal corpus to cryptic corpus?... Also, the normalization should be explained better (is crypto segment normalized to normal segment?). If so, caspase 3 is highly expressed in the cauda of crypto compared to normal so line 389 is confusing when mentions: “Conversely, caspase-3 expression showed an opposite trend with respect to Hsp70, showing a decrease compared to control…”. 
Line 485: [59[ should be changed to [59]
Line 489: (PFE) is an acronym for?

The manuscript has merit, and the reviewer considers it is suitable for publication after improvement.

Author Response

Dear Editor,

we appreciated the evaluation of the reviewers #1 and #2 in respect to our manuscript entitled “Aquaporins Are Differentially Regulated in Canine Cryptorchid Epididymis”.

In this regard, Reviewer’s comments have allowed us to significantly improve the manuscript: a) specifying some details on AQPs immunohistochemistry data, b) adding negative control in western blotting; c) adding more detailed insight into the contribution of AQPs to the dog reproductive biology, d) modifying the discussion to comment exclusively the results obtained.

Please find below a point-by-point reply to the Reviewers’ comments (reported in bold).

As requested, we used for the revision the file submitted first time and we applied the mode of “Track Changes”. Major changes are highlighted in yellow within the main body of the text.

We hope that the current revision could be suitable for the publication in Animals.

Regards,

Caterina Squillacioti, Nicola Mirabella, Giovanna Liguori, Giuseppe Germano and Alessandra Pelagalli

Comments of the Reviewer #2

GENERAL Comments and Suggestions for Authors

This manuscript described the expression patterns of specific aquaporins (7, 8 and 9) in the canine cryptorchid and normal epididymis using real time RT-PCR, immunohistochemistry (IHC) and western blot (WB) techniques.  No other publications have looked at aquaporins in these tissues in the canine cryptorchid model.  Thus, this provides some potentially interesting information and can be used in comparisons across species.  The results showed different patterns for the distribution and expression of the examined AQPs, upregulated in the caput and downregulated in the cauda region of the canine cryptorchid epididymis. These findings were associated together with the HSP70 and caspase-3 expressions to shed some light about the roles of these water channels in this pathophysiological condition.

I feel that the manuscript was well written, and that the methodology was well explained. While the discussion was relatively long, I feel that was essential to explain the results found, especially considering the number of proteins (AQP-7, -8, -9, HSP70 and caspase 3) that this study was investigating.

Answer to the Reviewer #2

General

We thank the reviewer # 2 for his/her appreciation of the quality of the manuscript in particular relative to the contribute due to the AQPs in canine cryptorchidism and to the possible roles of HSP70 and caspase-3 in course of this disease. Here there are reported point by point each modification of the text after the comments of the reviewer. The changes in the sentences of the manuscript were made by ticking the original text and by adding the new text. Every change was highlighted in yellow.

As suggested by the reviewer we tried to reduce the discussion eliminating several sentences regarding the cryptorchid condition and the role of epidydimal function and leaving only comments to explain discussion. However, as required by the reviewer#1 we needed to add some further information of AQPs expression in efferent ductules.

However, there are several concerns, both major and minor regarding the manuscript.

The major concerns are:

  • The IHC photographs presented in the reviewers file were of low quality, so it was difficult to discern what it was described.

Answer: we regret regarding the low quality of the IHC photographs included in the word file of the manuscript. We controlled the high quality of the figures, but we observed that their inclusion in the word file of the text reduced their quality so decided to submit them as separate files in the website used for the submission of the manuscript. We hope that in this way, the photographs could satisfy the judge of the reviewer in terms of quality and for the identification of cellular IR.

  • The IHC controls (positive and negative) are described in the Materials and Methods (lines 149-151), but the results of the negative controls should be included in the plates to avoid misinterpretation of background or unspecific immunolabeling (e.g. AQP-9 and nuclear labeling). In addition, the results/description of the IHC (specific cell location and /or tissue) for the AQP-7, -8 and -9 should be summarized in a table format as many of the papers cited by the authors including approximately quantification of stain in intensity and/or number of cells (%).

Answer: we thank the reviewer for his/her observation. We habitually perform positive and negative control to verify the immunolabeling especially for newly used antibody, but usually we do not include them in the figures of the manuscript but only when required..For this reason, we excuse for this carelessness. As suggested, we have included the negative controls as separate files in supplementary materials  Moreover, ss suggested by the reviewer, that we thank for his/her punctual observation we added two tables: 1) indicating the general expression of AQPs along the efferent ductules and the segments of epididymis of normal and cryptorchid dog; 2) indicating the cellular distribution of AQPs in the different epididymal tracts of normal and cryptorchid dog. 

  • There is no mention of negative controls described in the WB analysis, consequently neither in the results. Although the positive control is shown in the WB image, negative controls are necessary when the antibodies are not generated against the proteins of the specific species, in this case canine.

Answer: As answered for question (point 2), also for western blot technique, we performed experiments either use normal testis and kidney (positive control) or by omitting primary antibody (negative control), but in the figure we showed only positive control. As required by the reviewer, we have sent the western blotting containing the negative control (obtained by omitting the primary antibody against AQP7, AQP8 and AQP9) together with original blots as separate files in supplementary materials.

  • A morphological description of the lesions observed in the cryptorchid epididymis (epithelial lining and diameters of the ducts) would be appreciated (comparing D and F of the plates cauda epididymal segments).

Answer: we perfectly agree with the comment of the reviewer that we thank for helping us to improve the quality of the manuscript offering more precise information on the lesions observed in the epididymal tissue of the cryptorchid dog. The new sentences have been added in the initial part of the immunohistochemical results (results, page 6, lines 270-274).

Minor concerns:

  • The manuscript needs some minor language review.

Answer: we regret for the not excellent quality of the English language considering that prior of submission the entire manuscript was revised by using English Editing Service that certified the quality. After the revisions performed, the manuscript has been revised again prior to be re- submitted (R1). We hope that this new revision could be satisfactory for the reviewer.  

  • The use of “efferent duct” seems incorrect. There are several efferent “ductules” (ductuli efferentes) from the capital pole of the rete testis to the single tube forming the caput of the ductus epididymis. With that said, the size of the efferent ductules are considerably smaller in diameter than the caput epididymis and this is not perceived in the IHC photos (A versus B in figures 1, 2 and 3).

Answer: We thank the reviewer for his/her punctual suggestion. Accordingly, we modified efferent duct in “ductules” in the entire manuscript and added some sentences in the introduction to distinguish their organization in rodents and in other mammals (introduction, page 2, lines 54-66). The smaller size of efferent ducts compared to caput epididymis observed by the reviewer is due to the higher magnification of this region used for the figure to enhance the immunoreactivity.

3) The manuscript needs some editorial review:

Line 280: “B-D” instead of “B,D” or else describe C in the text.

Answer: We have modified the manuscript accordingly to your suggestion and clarified the points indicated (lines 305-307).

281: the symbol used in photograph D is not an asterisk (*) but a # (pound or number symbol).

Answer: We regret for the mistake. We modified the symbol in figure D

Line 293: same as above.

Answer: We regret for the mistake. We have modified the same as above

Line 390: Figure 5B should be Figure 6B. Also, the reviewer is confused about the comparison in the graph displayed in line 394 (Figure 6B) and the description presented in 389-390. Are the comparisons and the significant differences in the figure from normal caput to cryptic caput? Normal corpus to cryptic corpus?... Also, the normalization should be explained better (is crypto segment normalized to normal segment?). If so, caspase 3 is highly expressed in the cauda of crypto compared to normal so line 389 is confusing when mentions: “Conversely, caspase-3 expression showed an opposite trend with respect to Hsp70, showing a decrease compared to control…”.

Answer: We regret for the mistake regarding the figure 5B that we modified in Figure 6B. Regarding the comment to the Figure 6B, we agree perfectly with the comment of the reviewer and modified correctly the results of caspase 3 in the cauda segment (results, page 13, lines 437-438). The normalization of the cryptorchid cauda segment as well as of the remaining two regions was performed respect to each normal segment.

Line 485: [59[ should be changed to [59]

Answer: As suggested, we modified the text by adding correctly the square brackets

Line 489: (PFE) is an acronym for?

Answer: the acronym was modified in GFE that indicates good freezibility (discussion, page 16, line 551-552).